# Vaccine elicitation of HIV broadly neutralizing antibodies from engineered B cells

Deli Huang [1], Jenny Tuyet Tran[1], Alex Olson[1], Thomas Vollbrecht [2], Mary Tenuta [1], Mariia V. Guryleva[3,4], Roberta P. Fuller[1,5,6], Torben Schiffner[1,5,6], Justin R. Abadejos [1], Lauren Couvrette[1,7], Tanya R. Blane[1], Karen Saye[1,5,6], Wenjuan Li[1], Elise Landais[1,5], Alicia Gonzalez-Martin[8], William Schief [1,5,6,9], Ben Murrell [3✉], Dennis R. Burton [1,5,6,9✉], David Nemazee [1✉] & James E. Voss [1✉]

HIV broadly neutralizing antibodies (bnAbs) can suppress viremia and protect against HIV infection. However, their elicitation is made difficult by low frequencies of appropriate precursor B cell receptors and the complex maturation pathways required to generate bnAbs from these precursors. Antibody genes can be engineered into B cells for expression as both a functional antigen receptor on cell surfaces and as secreted antibody. Here, we show that HIV bnAb-engineered primary mouse B cells can be adoptively transferred and vaccinated in immunocompetent mice resulting in the expansion of durable bnAb memory and long-lived plasma cells. Somatic hypermutation after immunization indicates that engineered cells have the capacity to respond to an evolving pathogen. These results encourage further exploration of engineered B cell vaccines as a strategy for durable elicitation of HIV bnAbs to protect against infection and as a contributor to a functional HIV cure.

[1] Department of Immunology and Microbiology, The Scripps Research Institute, La Jolla, CA, USA. [2] Department of Medicine, The University of California San Diego, La Jolla, CA, USA. [3] Department of Microbiology, Tumor and Cell Biology, Karolinska Institutet, Stockholm, Sweden. [4] Faculty of Bioengineering and Bioinformatics, Moscow Lomonosov State University, Moscow, Russia. [5] Scripps Consortium for HIV/AIDS Vaccine Development (CHAVD), The Scripps Research Institute, La Jolla, CA, USA. [6] IAVI Neutralizing Antibody Center (IAVI), The Scripps Research Institute, La Jolla, CA, USA. [7] Faculty of Science, University of Ottawa, Ottawa, Canada. [8] Department of Biochemistry, Universidad Autónoma de Madrid (UAM) and Instituto de Investigaciones Biomédicas Alberto Sols (CSIC-UAM), Madrid, Spain. [9] Ragon Institute of Massachusetts General Hospital, Massachusetts Institute of Technology, and Harvard, Cambridge, MA, USA. ✉email: benjamin.murrell@ki.se; burton@scripps.edu; nemazee@scripps.edu; jvoss@scripps.edu

Critical support for efforts to design an HIV vaccine that would elicit broadly neutralizing antibodies (bnAbs) comes from experiments that show such antibodies are capable of providing sterilizing immunity against clinically relevant (so-called Tier 2) virus challenge in macaque and humanized mouse models[1–3]. Genetic limitations imposed by the human repertoire of B cell antigen receptors (BCRs) from which such antibodies must arise, has however significantly impeded these efforts. Characterization of a variety of bnAbs discovered in chronically infected patients shows that they generally derive from B cell precursors with uncommon antigen receptor features, such as long third heavy chain (HC) complementarity determining regions (CDRH3s), and then require extensive somatic hypermutation to facilitate broad neutralization[3,4]. Mature HIV bnAbs and other desirable antibody genes can be directly added to the repertoire by engineering them into the genomes of ex vivo activated primary B cells for expression as functional BCRs using endogenous HC constant genes[5–7]. As such, engineered BCRs can undergo class switching for eventual secretion as protective antibodies from plasma cells. B cells engineered in this way have been shown to confer protective levels of pathogen-specific antibody in vivo for several weeks following adoptive transfer into immunocompromised mice[7], or for several days following transfer into immunocompetent wild type (WT) mice[6,7]. Vaccine elicitation of durable HIV bnAb responses from engineered B cells in immunocompetent hosts would be an attractive contributor to a functional HIV cure, given that bnAbs administered passively in the context of infection have been shown to suppress viremia, kill infected cells, and enhance host immunity[8–10]. Here, we show that HIV bnAb-engineered primary B cells can be expanded and affinity-matured in vivo through vaccination, resulting in durable bnAb memory and long-lived plasma cells in wild-type mice.

## Results

**Modified primary B cells express the HIV VRC01 bnAb as antigen receptor.** Wild-type (WT) C57BL/6J (or B6) mice are a useful model in which to explore vaccine elicitation of HIV bnAbs from engineered cells because of similarities between the mouse and human humoral immune systems, and because such bnAbs cannot be elicited from the natural repertoire[11–13]. We therefore developed methods to knock the VRC01 HIV bnAb into mature primary C57BL/6J B cells. We chose VRC01 because this prototype CD4 receptor binding-site bnAb has been extensively tested in the clinic for its ability to suppress viremia in patients and prevent infection after administration as a recombinant monoclonal antibody[14–16] (clinicaltrials.gov NCT02568215, NCT02716675). In our first genome editing strategy, donor DNA carrying a V-gene promoter, the VRC01 HIV bnAb[17,18] light chain (LC), HC variable region (VDJ), and constant gene splice donor site, are introduced into a CRISPR-cas9 DNA break-site in the *IgH-J4* gene using homology-directed repair (HDR) in activated cells. Here, the inserted VRC01 genes are expressed from a single mRNA, transcript spliced to downstream endogenous *IgH* constant genes. A P2A self-cleaving peptide sequence downstream of the VRC01 mouse kappa (κ) constant gene separates the light and HCs, allowing them to pair and form a functional cell surface-expressed BCR (*H-targeting*) (Fig. 1a). A second engineering strategy was also employed, in which the VRC01 heavy and κ variable regions were targeted separately to their endogenous loci for expression from V-gene promoter-controlled transcripts spliced to cell-native H or κ constant genes (*H + κ targeting*) (Fig. 1b). Targeting success in primary B cells was monitored using HIV envelope glycoprotein-based flow cytometry probes that specifically recognize VRC01: eOD-GT8 is an

'engineered outer domain' of gp120 that presents the CD4-binding site and has a $K_D = 16$ pM monovalent affinity to VRC01; KO11 carries a single amino acid substitution that knocks out VRC01 binding[19,20]. Targeting by *H* or *H + κ* strategies routinely resulted in engineering efficiencies of 10% and 1%, respectively when plasmid donor DNA and CRISPR-cas9 ribonucleoprotein (RNP) was delivered to LPS-activated cells using electroporation (Fig. 1c, Supplementary Fig. 1). While less toxicity was observed when adeno-associated viral (AAV) vector donor DNA was transduced into cells after RNP electroporation (Supplementary Fig. 1), we preferred plasmid donors as targeting efficiencies were comparable, and because this format allowed for quick and inexpensive development of a large number of donor DNAs for testing. Cutting efficiencies of the *IgH-J4* and *IgK-J5* RNPs were 80 and 55% by TIDE analysis[21] (Supplementary Fig. 1b). Off-target repair of these DNA breaks should either be inert or generate BCR knockouts which will lead to cell apoptosis. Translocation of the telomeric ends of mouse chromosomes 12 and 6 (involving the two cuts generated in $H + k$ targeted cells) would also lead to a loss of BCR expression and cell apoptosis.

**Ex vivo targeted B cells return to rest as self-tolerant memory in vivo.** These engineering strategies can result in the expression of self-reactive BCRs due to pairing of engineered Ig chains with those endogenously produced by the targeted cell[5–7]. This is especially true for *H-targeted* cells which present native LCs on the surface of at least 34% of VRC01-expressing cells (Supplementary Fig. 2a). While tolerance mechanisms in the periphery of mice and humans generally ensure that autoreactive B cells are non-functional[22–25], we sought to ensure that this would still be the case for engineered B cells which require an ex vivo lipopolysaccharide (LPS) activation step in order to achieve efficient HDR based genome editing. We therefore transferred WT untouched, or ex vivo LPS-activated B cells into transgenic mice expressing an Igκ chain-reactive super-antigen on the surface of hepatocytes (pAlb mice)[26]. In this context, all donor Igκ+ B cells would be autoreactive. After 28 days in this host, both the untouched and LPS-activated Igκ+ cells were deleted suggesting that auto-reactive B cells generated during the engineering step should remain subject to peripheral tolerance mechanisms in vivo (Supplementary Fig. 2b, c).

B cells purified from the spleens of WT donor mice by negative selection were then activated with LPS, targeted, and transferred into WT recipients to assess their status after 14 days in vivo. Untouched donor B cells that were directly transferred into control mice had low frequencies of memory B cells (MBCs), whereas 70-80% of ex vivo targeted cells acquired a germinal center (GC)-independent (CD73−) memory phenotype[27,28] (Fig. 1d–f, Supplementary Fig. 3). The serendipitous effect of generating memory after 14 days in vivo as a result of ex vivo LPS activation for genome editing suggests engineered B cells should be optimally poised for vaccination at this timepoint.

**Durable VRC01 serum titers can be elicited from H-targeted cells by vaccination.** After targeting using either the *H* or $H + κ$ engineering strategies, 40,000 VRC01 BCR-expressing cells were engrafted per animal into WT recipient mice. This corresponds to ~3 in 100,000 B cells present after parking for 14 days in vivo (Supplementary Fig. 3b). Mice with either mock-targeted, $H + K$ *targeted*, or *H-targeted* B cells were primed 14 days post cell transfer (w0) with MD39-ferritin, a soluble stabilized HIV envelope native trimer multivalently presented on a ferritin-based nanoparticle. MD39 has a monomeric affinity for VRC01 of $K_D = 124$ nM[29,30]. Mice were boosted 6 weeks (w6) after a prime. Three mice engrafted with *H-targeted* cells involved in a

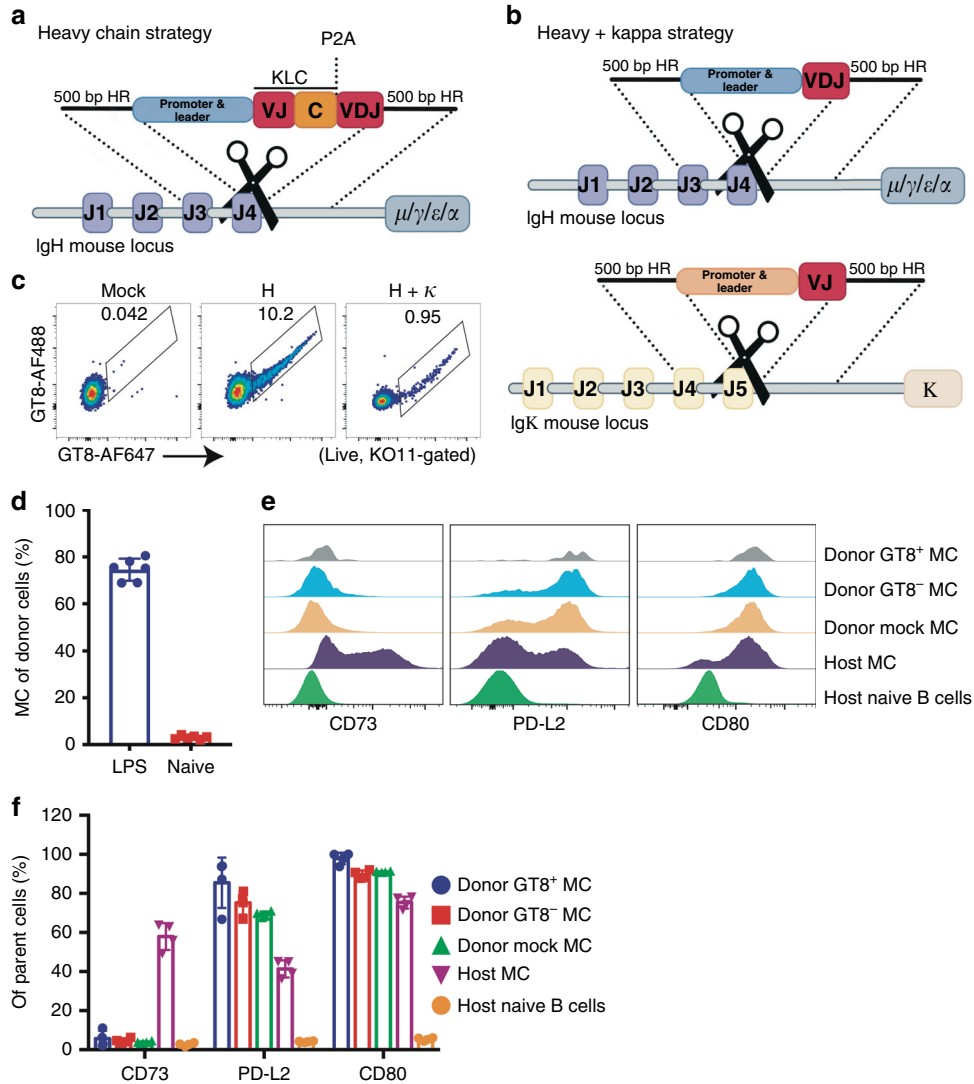

**Fig. 1 Engineering primary B cells and adoptive transfer of cells to WT mice. a** Targeting antibody genes to the mouse heavy chain (HC) locus (*H-targeting*). Donor DNA encoding (1) a HC V-gene promoter, (2) the VRC01 κ chain (with mouse constant gene and P2A peptide), (3) the VRC01 HC variable (VDJ) region and donor splice site is inserted at a CRISPR-Cas9 cut site in JH4 for expression as a functional antigen receptor using endogenous downstream HC constant genes. Regions of homology (HR) flanking bnAb donor-DNA allow for its incorporation at the DNA break-site by homology-directed repair (HDR). **b** Targeting antibody genes to the mouse heavy and kappa loci (*H + κ targeting*). To engineer the *IgH* locus, donor DNA encoding (1) a HC V-gene promoter (2) VRC01 VDJ gene, and constant gene donor splice site is inserted as above. To engineer the Igκ locus, donor DNA encoding (1) a κ V-gene promoter, (2) VRC01 κ variable (VJ) region and constant gene donor splice site is likewise inserted into a CRISPR-Cas9 cut site in Jκ5, for expression of VRC01 H and κ chains from their endogenous loci spliced to cell-native constant genes. **c** Targeting efficiency. Successfully targeted B cells expressing VRC01 as cell surface antigen receptor were detected as live, single, KO11⁻,eOD-GT8-AF647⁺, and eOD-GT8-AF488⁺ cells by flow cytometry. eOD-GT8 double-positive cells are shown for LPS activated (mock), *H + κ*, and *H-targeted* B cell cultures. **d** LPS-activated donor cells acquire a memory phenotype in vivo after adoptive transfer. Non-engineered primary B cells were either directly transferred or cultured for 48 h in LPS before adoptive transfer into host mice. The fractions of donor (CD45.1⁺) cells that showed a memory cell (MC) phenotype after 14 d in vivo are shown for n = 6 mice in each group. Memory B cells are gated as live CD19⁺CD38⁺GL7⁻IgD⁻. **e** Engineered cells become germinal center independent (CD73⁻) memory in vivo. *H-targeted* cells were adoptively transferred into host mice. 14 days later, successfully engineered (GT8⁺) cells were analyzed by flow cytometry. Host naïve and memory B cell populations are compared for their expression of CD73, PD-L2, and CD80 memory markers. **f** Quantitation of B cells gated as in (**e**). The fraction of successfully targeted cells with the indicated cell surface memory cell markers are given for n = 4 engrafted mice. These are compared with unsuccessfully engineered cells in the targeted population (GT8⁻), mock engineered, or host cell controls. Data are presented as mean value ± SD in Figure d and f.

longer-term study were further boosted 13 and 32 weeks after the prime (Fig. 2a). Total antigen-specific and engineered antibody responses were tracked in the serum by ELISA and using virus neutralization assays. Mice that received mock engineered or *H + κ targeted* cells elicited the lowest total antigen-specific and VRC01-competitive antibody responses. Mice that received *H-targeted* cells showed on average 10-fold higher total antigen-

specific titers one week after the first boost and this was accompanied by significantly higher levels of VRC01-competitive antibody in the serum (Fig. 2b, c). Because some VRC01-competing antibody responses could be elicited in control mice with mock-targeted cells, VRC01 in the serum of mice containing *H-targeted* cells was also measured by P2A ELISA, as VRC01 κ chains expressed from these cells are tagged with a P2A

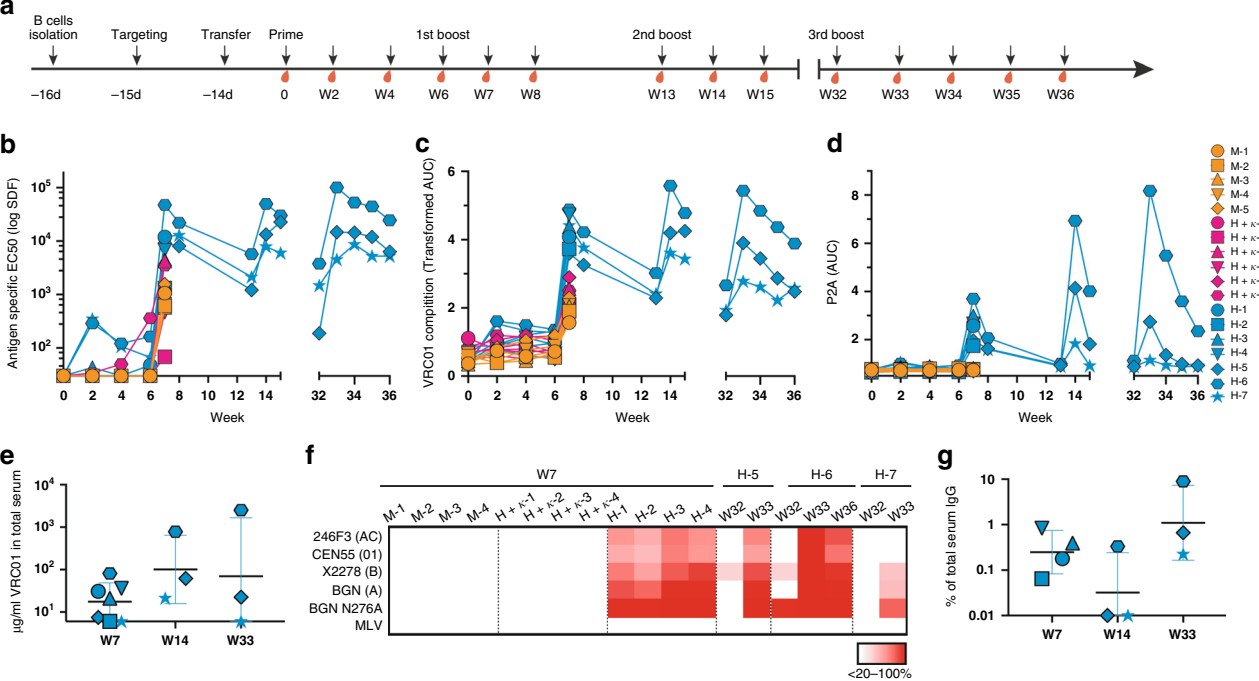

**Fig. 2 Serological analysis of VRC01 responses after vaccination. a** Engineered B cell vaccine experimental design. Time course of B cell engineering, cell transfer, immunization, and blood sampling in WT C57BL/6J mice. **b–d** Serum antibody responses after MD39-ferritin immunization in mice that received mock targeted, *H-targeted*, or *H + κ targeted* B cells. Titers (*y*-axis) are given for the indicated timepoints (*x*-axis) and mice (legend): **b** Total (host + donor) IgG response to the immunogen, given as serum dilution factor (SDF) EC50s; **c** VRC01-competetive antibody responses given as the transformed area under the curve (AUC) and; **d** Serum IgG carrying the P2A tag on engineered L-chains, given as area under the curve (AUC). **e** μg/ml VRC01 in the serum. The quantity of VRC01 in the indicated serum samples at the indicated timepoints was calculated using a recombinant mouse VRC01-P2A-tagged IgG standard. $n = 3$-4 mice per group in $n = 2$ experiments. **f** Serum neutralization of HIV. IgG purified from the serum of the indicated mice at the indicated timepoints was tested for its ability to neutralize HIV pseudoviruses using the TZM-bl assay. Percent neutralization of virus achieved by 200 μg/ml of IgG is given as a heat map for several tier-2 viruses from different clades (BGN = BG505) when neutralization decreases with IgG concentration. **g** VRC01 antibody as a fraction of the total serum IgG. The fraction of total serum IgG with functional VRC01 activity was assessed for the indicated mice at the indicated timepoints (weeks post-prime) by comparing the BG505-N276A neutralization IC50 values for these samples with the recombinant mouse VRC01-P2A-tagged IgG. $n = 3$-4 mice per group in $n = 2$ experiments. Data are presented as mean value ± SD in (**e**) and (**g**).

self-cleaving peptide at the C terminus. Mice with *H-targeted* cells elicited P2A titers after each boost (Fig. 2d, Supplementary Fig. 4). The highest responder (H-6), produced ~2.4 mg/ml of VRC01 in the serum one week following the third boost calculated using a recombinant LC P2A-tagged VRC01 mouse IgG1 standard (Fig. 2e, Supplementary Fig. 4). IgG purified from sera of these mice also showed VRC01-like neutralization of HIV pseudoviruses from different clades that bear hallmarks of typical circulating viruses, such as relative neutralization resistance[31] (Fig. 2f, Supplementary Fig. 5). Because no neutralizing responses could be elicited to a strain particularly sensitive to VRC01 neutralization (BG505-N276A) in mice with mock-targeted cells, IC50s for this pseudovirus were used to quantify functional VRC01 as a fraction of total serum IgG in responding mice using the recombinant LC P2A-tagged VRC01 mouse IgG1 standard. In the best responder (H-6), serum VRC01 constituted ~0.15% of the total IgG even after a 5-month rest period (w32) and this was boosted to ~10% of the total IgG by one week after the third boost (Fig. 2g, Supplementary Fig. 5). We conclude that *H-targeted* engineered B cells can generate durable memory and serum antibody in the immunocompetent B6 mouse model. Further investigation is required to understand why *H + κ targeted* cells failed to respond detectably to vaccination.

### Immunized VRC01-B cells mature into GC-dependent memory and plasma cells. Consistent with the serum analysis, two weeks after the first boost, only mice that received *H-targeted* cells

showed high frequencies of VRC01-engineered GC B cells (Fig. 3a-c), splenic and bone marrow plasma cells (Fig. 3d, e) and MCs (Fig. 3f). Engineered cells in mice from this group were expanded 20–70-fold during the prime and boost (Fig. 3g) and VRC01-expressing memory (GT8++, KO11−, donor+) cells became mostly CD73+, implying their entry into GCs after vaccination, unlike GT8− donor cells, which remained CD73− after LPS treatment (Fig. 3h, i).

Repeat experiments and variations on the original protocol also successfully elicited VRC01 memory responses in other groups of mice with *H-targeted* cells (Supplementary Table 1). Variations included: (1) FACS enrichment of only antigen-specific engineered cells before transfer; (2) ex vivo activation of B cells using CD40L, IL-4, and other MyD88 inducers such as CpG DNA (rather than LPS); (3) alternative immunization strategies in which engineered cells were primed coincident with transfer into mice with pre-existing immunogen-specific T cell help, and; (4) immunization using different adjuvants.

### Coding mutations are selected in VRC01 genes after vaccination. Engineered immunoglobulin gene repertoires from the spleen (memory) and bone marrow (long-lived plasma cell) compartments were sequenced in several mice with *H-targeted* cells 35 days after their final boosts. Repertoires mostly conserved the original VRC01 amino acid sequence, however an abundance of variants with coding changes were present, including some highly mutated lineages emerging in the memory compartment

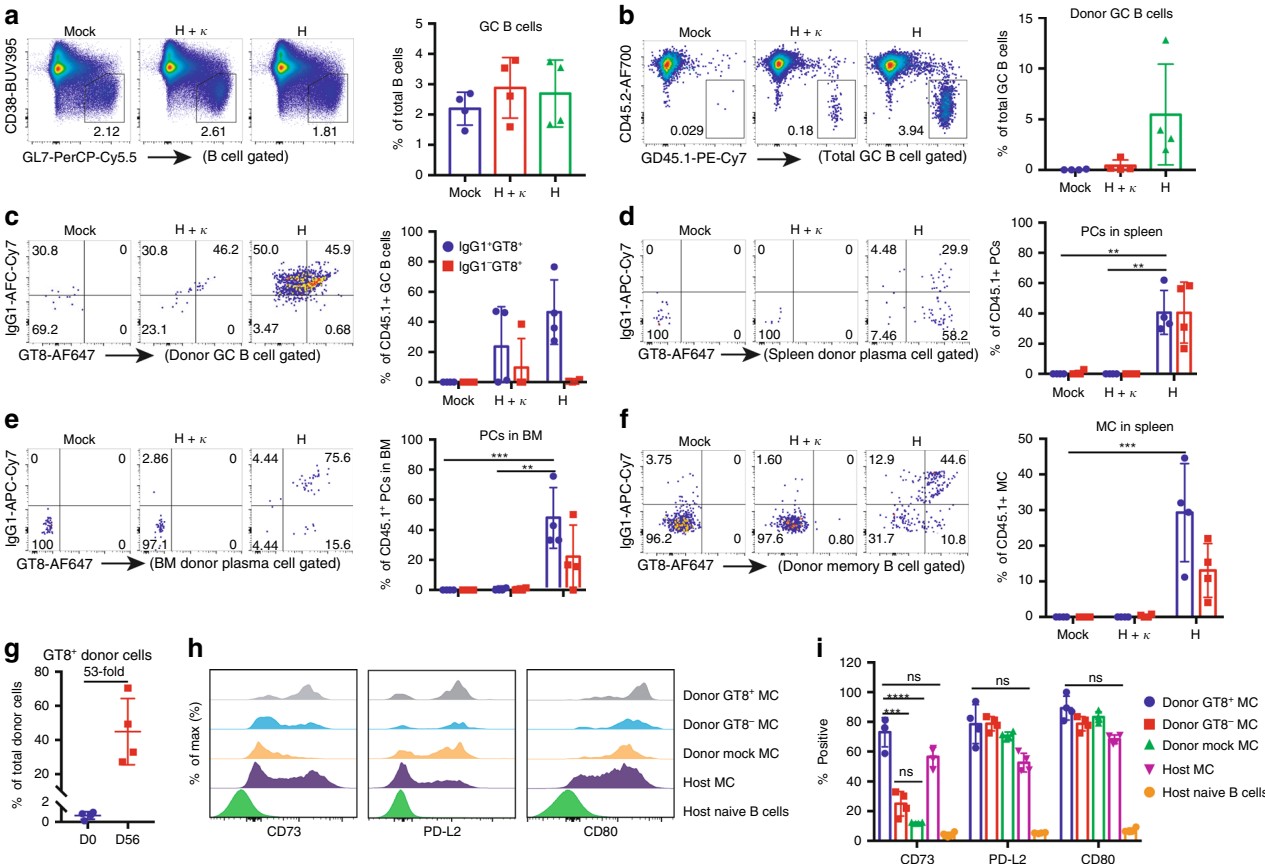

**Fig. 3 Flow cytometric analysis of antigen-matured engineered B cells.** Host mice receiving LPS activated only (Mock), $H + \kappa$, or $H$-targeted B cells were analyzed 14 d after the first boost (w8). $n = 4$ mice in each group. **a–c**, Germinal center (GC) B cells. **a** Representative flow cytometry and statistical analysis of total GC B cells pre-gated as live, CD19+ singlets. **b** CD45.1+ (donor) GC B cells. **c** Fraction of cells gated in (**b**) which were class-switched IgG1+ and GT8++KO11− (engineered VRC01-expressing). **d–e** Plasma cells (PCs). Frequency of class-switched and antigen-specific PCs in the spleen (**d**) and bone marrow (**e**) as a fraction of total donor PCs. GT8++KO11−CD45.1+CD45.2−IgG+ PCs were analyzed by intracellular staining after the permeabilization of surfaced-stained cells. Plasma cells are gated as F4/80−IgD−TCRβ−CD138+Sca-1+. **f** Memory cells (MCs). Enumeration of IgG1+GT8++KO11−, engineered MCs (sIgD−CD45.1+CD45.2−CD38+GL7−). **g** Vaccine-induced expansion of engineered cells. GT8++KO11−CD45.1+CD45.2− cell expansion at w8 compared with w0 as a percentage of the total donor cells. **h** Engineered memory cells have become germinal center dependent (CD73+) after vaccination. GT8+ or GT8− donor memory cells were compared with host naïve and memory B cell controls for expression of the indicated memory cell surface markers. **i** Quantification of B cells memory subtypes gated as in (**h**). **a–i** Bars represent mean ± SD for all data points in each group. *P < 0.05; **P < 0.01; ns, not significant; unpaired 2-tailed T test.

(Fig. 4a). While engineered cells were mostly IgM at the time of transfer, the VRC01 repertoires after vaccination were almost completely class switched with IgG1 and IgG2 isoforms dominating as expected (Fig. 4a, b). Mutations in the post-vaccinated engineered repertoire were few for IgM and IgG3, while the majority of sequences of all other isotypes were mutated, with some IgG1 sequences showing as high as 4% divergence from the original donor DNA sequence (Fig. 4b, Supplementary Fig. 6). Coding mutations were observed across the entire length of the engineered VRC01 LC and HC variable regions, especially within the κ constant gene, which is not normally in the path of the mutator (Fig. 4c, Supplementary Fig. 7a). Structural modeling shows that these κ constant mutations mostly occur in solvent-exposed loops connecting β-strands of the immunoglobulin-fold and would be functionally innocuous (Supplementary Fig. 7b). Coding changes in the P2A peptide were strikingly absent as this sequence must be highly conserved in order to express a functional BCR in engineered cells. Identical mutations could be observed across compartments and animals, and some coding changes appeared enriched in one compartment over the other within the same animal (Fig. 4c). The HC CDR2 region of VRC01, which forms the primary interaction surface of this

antibody with the virus[17], showed coding changes after vaccination (Fig. 4d) indicating that engineered responses generate some variation in antigen-binding specificity which could be advantageous against highly diverse viral reservoirs and confirming the ability of engineered B cells to undergo somatic mutation and affinity maturation.

In the first demonstration of antibody activity, Behring and Kitasato showed in 1890 that serum from immunized mice could protect naive ones from lethal challenge of Diptheria and Tetanus toxins. We show here, and in the complementary study by Nahmed et al.[32], that it is possible to passively transfer genetic information to the adaptive immune system that facilitates a high affinity, highly evolved, yet further evolvable, HIV bnAb response —in effect, demonstrating that one can program immune memory. Thus, we envision that our study represents a new phase in the development of passive immunity, started with the discovery of the antibody itself.

## Methods

**Cell culture.** 293T cells for HIV pseudovirus production were obtained from the American Type Culture Collection (ATCC, #CRL-3216) and cultured in DMEM (Invitrogen, #10313021) supplemented with; 10% fetal bovine serum (Invitrogen,

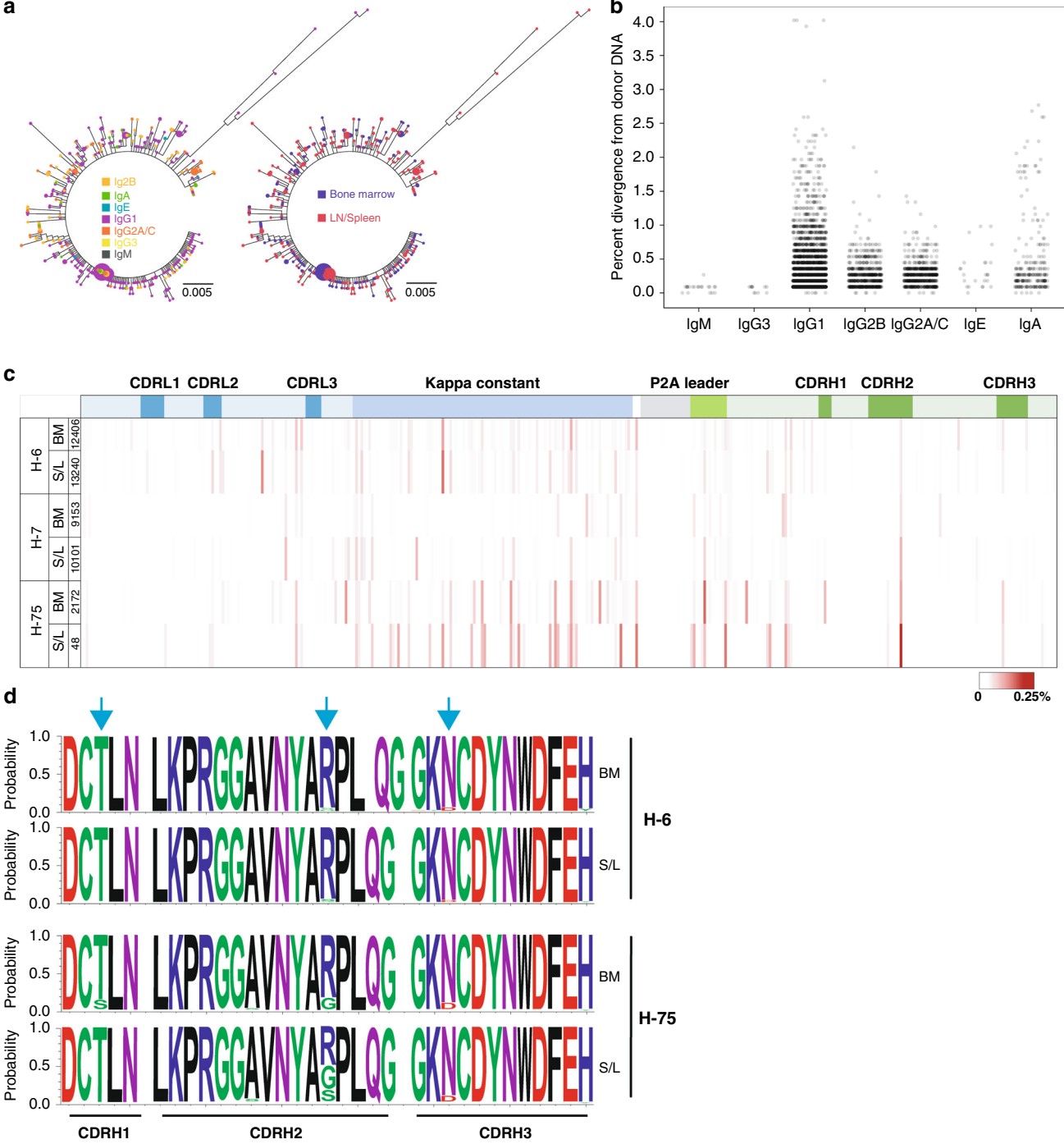

**Fig. 4 Engineered antibody repertoire in vaccinated mice. a** Diversity of the engineered repertoire after immunization. Relatedness of VRC01 clonotypes by isotype (left), and compartment (right), in one representative animal (H-7) 5 weeks after the final boost. **b** Somatic hypermutation by isotype. Percent divergence from donor DNA nucleotide sequence is shown for combined engineered repertoires from all compartments and animals sequenced by isotype. Fewer engineered IgM or IgG3 sequences were obtained and these were mostly unmutated (<5% of sequences), while IgG1, G2B, G2A/C, E, and A were significantly mutated (>68% of sequences, reaching up to 4% mutated) (Supplementary Fig. 6). Fig. S6). **c** Mutational hotspots in the engineered VRC01 gene. The frequency of amino acid changes at each residue position across the *H-targeted* VRC01 gene is shown as a percentage of the total sequences obtained for each dataset. Six datasets are shown which are derived from both memory and plasma cell compartments from the three indicated mice. Specific coding changes across the length of the gene are given in Fig. S6 from one representative mouse (H-6). **d** Antigen-binding properties are diversified in the engineered repertoire. The fraction of sequences with coding changes from memory (S/L) or plasma cell (BM) compartments are shown as sequence logos for the CDRH regions in the indicated mice. Blue arrows indicate amino acid positions undergoing diversification.

#26140-079), 2 mM L-glutamine (Invitrogen, #2503008), 100 units of Penicillin and 0.1 mg/ml of Streptomycin (Invitrogen, #15140122). TZM-bl cells for virus neutralization assays were obtained from the National Institutes of Health (NIH) AIDS Reagent Program (#8129) and cultured in DMEM supplemented with FBS (10%), 100 units of Penicillin and 0.1 mg/ml of Streptomycin. FreeStyle™ 293F cells for protein production were obtained from Life Technologies (#R79007) and cultured in FreeStyle™ 293 Expression Medium (Life Technologies, #12338018). Chemically competent DH5a E. coli for plasmid propagation were purchased from NEB (#C2987H).

**Mice.** Mouse studies were approved and carried out in accordance with protocols provided to the Institutional Animal Care and Use Committee (IACUC) at Scripps Research (La Jolla, CA) under approval number 18-0004. The mice were housed, immunized, and bled at Scripps in compliance with the Animal Welfare Act and other federal statutes and regulations relating to animals in adherence to the Guide for the Care and Use of Laboratory Animals (National Research Council, 1996). All experiments were carried out in wild type or pAlb 3-month-old female C57BL/6J (CD45.1/Ly5.1 or CD45.2/Ly5.2) mice bred at Scripps Research Division of Animal Resources Facility (DAR). Mice were housed at ambient temperature and humidity on a 12 h light cycle. Not more than 5 mice or less than 2 mice were housed together. All procedures were performed on animals anesthetized using isoflurane.

**Cas9-RNP selection.** gRNAs targeting the IgH-J4 and IgK-J5 region of the reference C57BL/6J genome (GRCm38) were designed using the GPP webtool (https://portals.broadinstitute.org/gpp/public/analysis-tools/sgrna-design). The gRNAs were synthesized by IDT (as crRNAs). CRISPR-Cas9 tracrRNA and HiFi Cas9 nuclease V3 were purchased from IDT (#1072534, #1081061). Cas9:crRNA:tracrRNA complexes were made according to the manufacturer's instructions. The targeting efficiency of different gRNAs was directly assessed by VRC01 engineering efficiency in LPS-activated primary B-cells (described below) and the best gRNAs were chosen for IgH-J4 and Igk-J5 targeting. Individual gRNA targeting efficiencies were assessed by transfecting each one into cells as RNPs. After 48 h of culture, gDNA was purified (Qiagen 69504) and amplicons of the target cut sites generated by PCR (H target PCR primers: forward 5′-tcttgtgtgacaccaggaattggcat-3′, reverse 5′-cctccagccatagggattgttttagca-3′. k target PCR primers: forward 5′-cttgtgtgaatttgtga-catttttggct-3′, reverse: 5′-acgtctagaagaccacgctacctgca-3′). PCR products were purified (Qiagen, 28004) and Sanger sequenced (Eton Bioscience) using both forward and reverse primers (H target sequencing primers: forward 5′-ggaaaaatccactattgt-gattactatgct -3′, reverse 5′-cccatcatccagggactccaccaaca-3′. k target sequencing primers: forward 5′-agaaaagagtcagtgtgaaagctgagcga-3′, reverse: 5′-ttctccgatccaatctcttggatggtg-3′). Chromatograms were uploaded into TIDE for INDEL analysis as instructed by the developer (http://shinyapps.datacurators.nl/tide/)[21].

**HIV bnAb donor DNA preparation.** H-targeting VRC01 donor DNA was designed as follows from 5′ to 3′: (1) 498-basepair (bp) homology arm upstream of the mouse IgH-J4 (in germline configuration). (2) The mouse IgHV1−82 HC promoter region (entire 5′-UTR). (3) The mouse IgKV4-53 leader (intron removed) and human VRC01 light-chain variable gene followed by the mouse κ constant region without stop codon. (4) A GSG linker followed by the P2A self-cleaving peptide sequence. (5) The mouse IgHV1−82 leader lacking the intron followed by the human VRC01 heavy-chain variable gene and constant gene donor splice site. (6) A 500-bp region of DNA homologous to the IgH-J4 intron beginning after JH4. The homology regions and V-gene promoter were amplified from B6 gDNA. The VRC01 LC-P2A-HC VDJ gene was synthesized (Geneart). H + κ targeting donor plasmids were designed as follows: The H-targeting plasmid was modified by the deletion of the LC and P2A peptide and adding in the IgHV1-82 intron. The LC donor DNA was designed as follow from 5′ to 3′: (1) 510 bp homology arm upstream of the mouse IgK-J5; (2) the mouse IgKV4-53 LC promoter (entire 5′ UTR), leader and V-gene intron; (3) VRC01 VJ and constant gene donor splice site (4) a 510 bp region downstream of Jκ5. All fragments were assembled using Gibson assembly (NEB, #E5510S) according to the manufacturer's instructions into the PICOZ (PMID: 31124712) carrier vector synthesized from IDT and transformed into DH5a E. coli. (NEB, C2987H). Single colonies from bacteria plated on agar containing Zeocin were cultured; plasmids were isolated (Qiagen, #27106) and sequenced (Eton Biosciences) using several primers to generate high-quality coverage of the entire donor DNA region. Donor DNA plasmids were purified using EndoFree plasmid maxi kit (Qiagen, #12362) for engineering experiments.

**B-cell activation and engineering.** All work was conducted under sterile conditions. B-cells isolated from the spleens of CD45.1 or CD45.2 congenic mice using negative selection (Miltenyi, #130-090-862) were cultured in RPMI-1640 (Invitrogen, #21870076) supplemented with 1X NEAA (Invitrogen, #11140050), 1X sodium pyruvate (Invitrogen, #11360070), 55 µM 2-Me (Invitrogen, #21985023), 10% FBS (Invitrogen, #26140-079) and either (1) 50 µg/ml LPS (Sigma, #L2880-100MG) or (2) CpG ODN 2006 at 1 µM (Synthesized from IDT), IL-4 at 50 ng/ml (Biolegend, #574306), anti-CD40 at 5 µg/ml (Thermo, #16-0401-86) for 30 h. The Alt-R CRISPR-Cas9 system components (IDT) were used to make Cas9-RNP complexes according to the manufacturer's instructions. gRNA sequences for H or

κ targeting were 5′-gagaggccattcttacctg-3′ and 5′-ttacgtttcagctccagct-3′ respectively. Activated cells were washed three times with 1X DPBS (Invitrogen, #14190-144) and resuspended at 5 million cells/ 100 µl in Neon R buffer (Invitrogen, #MPK10096). For H-targeting, 5 µl H-targeting Cas9-RNP complexes, 2.16 µl 100 µM CRISPR electroporation enhancer (IDT), and 20 µg donor DNA (4 µl) were mixed and added to the resuspended cells. For H + κ -targeting, 5 µl each of H- and κ- targeting Cas9-RNP complexes, 4.32 µl 100 µM CRISPR electroporation enhancer (IDT), and 15 µg each of HC and LC donor DNAs were mixed and added to the resuspended cells. The cells were electroporated using a 100 µl tip (Invitrogen, MPK10096) with 1650 V, 20 ms, 1 pulse. Nucleofected cells were immediately cultured in media without antibiotics or cell activation components for 1 h before these components were added for further ex vivo culture. When donor DNA was transduced into cells as AAV, VRC01 AAV2 donors produced by Vector Biolabs using the same H-targeting design were used. Cells were at a concentration of 1 million in 10 µl RPMI-1640 serum-free media were mixed with equal volumes AAV at an MOI of $2 \times 10^5$ for 1 h, then electroporated as described above.

**Adoptive transfer of engineered cells.** Eighteen hours after transfection, activated mock or engineered cells were washed four times with DPBS (Invitrogen, #14190-144) for adoptive transfer into recipient cells. Some cells were further cultured to confirm the quantity of successfully targeted cells by flow cytometry 48-h post-transfection. VRC01-engineered B cells were identified by staining for viable (propidium iodide negative), eOD-GT8+, eOD-GT8-KO11− cells. In some experiments (Table S1), successfully targeted cells were enriched by FACS before transfer 18-h post-transfection by gating the live, GT8++KO11− population. To prepare eOD-GT8 probes, 1.1 µg of biotinylated GT8-monomer (biotinylated using the EZ-link NHS-Biotin from Thermofisher) was mixed with 0.88 µg streptavidin-488/647.

**Immunization of rested cells.** 4 million H + κ targeted VRC01 cells or 1 million H-targeted VRC01 cells (corresponding to 40,000 VRC01-expressing engineered cells by either strategy) were retro-orbitally transferred into recipient mice. These mice were rested for 14 d before immunization with a 200 µl mixture of 20 µg immunogen and either Ribi adjuvant (according to the manufacturer's instructions, Sigma, #S6322-1VL) or 5 µg of IscoMPLA (kindly provided by the lab of Darrell Irvine at MIT) in DPBS administered by i.p. injection. Mice were boosted with the same immunogen 6, 13, and 32 weeks after priming. Serum samples were collected through ocular bleeding of mice at the time intervals indicated in Fig. 2. All procedures were done on isoflurane-anesthetized mice.

**Immunization of T-primed cells.** Recipient mice were i.p. immunized with 200 µl mixture of 10 µg of either the lumazine synthase or ferritin base in DPBS and Ribi adjuvant. 7 days later, targeted cells containing 65,000 VRC01-expressing engineered cells were retro-orbitally transferred into pre-primed host mice. Lumazine synthase pre-primed mice were immunized with 20 µg eOD-GT8-60mer and ferritin pre-primed mice were given 20 µg MD39-ferritin immunogen in Ribi adjuvant at the same time. For the FACs enriched group, 300,000 sorted antigen-specific cells were transferred into recipient mice following the same manner of T-primed immunization. All procedures were done on isoflurane-anesthetized mice.

**Flow cytometry analysis and single-cell sorting.** Spleen suspensions were generated by smashing the spleen between frosted glass slides. Bone marrow was released from tissue-free tibia and femurs by a mortar-and pestle. Red blood cells (RBCs) were lysed with ammonium chloride (0.83%) before filtering cells through a 40 µM cell strainer to generate single-cell suspensions. Fc Blocker (homemade mAb 2.4g2) was added to single-cell suspensions at 0.5 µg per $10^6$ cells before antibody staining. For antigen-specific analysis, biotinylated AviTagged GT8 monomers were pre-complexed for 30 min to Streptavidin-AF488 (Thermo, #S32354) or Streptavidin-AF647 (Thermo, #S32357); GT8-KO11 was complexed to streptavidin-BV421 (BD, #BDB563259). Fluorophore-conjugated antibodies CD45.1 (Biolegend, #110728), CD45.2 (Biolegend, #109806), GL7 (Biolegend, #144608, #144610) TCRb (Biolegend, #109228), F4/80 (Biolegend, #123128), Ter119 (Biolegend, #116228), CD38 (Biolegend, #102718), IgD (Biolegend, #405710), IgM (Biolegend, #406512), IgG1 (Biolegend, #406620), CD138 (Biolegend, #142504), Sca-1 (Ly6A/E) (Biolegend, #122512) and CD19 (Biolegend, #152408), CD80 (Biolegend, #104712), CD73 (Biolegend, #127210), PD-L2 (Biolegend, #107216), anti-mouse Kappa (Clone 187.1 AF647), Anti mouse Lambda (Biolegend, #407306) were used to define different cellular populations. CD45.1 and CD45.2 mAbs were used to distinguish host and transferred B-cells. IgG1 staining was used to identify class-switched B-cells. VRC01 expressing memory cells (MCs) were gated as live GT8+KO11−CD19+CD38highsIgD−GL7−; Germinal center (GC) B-cells were gated as live CD19+ CD38−GL7+; Plasma cells (PCs) were gated as CD138+ Sca1+TCRb−Ter119−sIgD−sIgM−GL7−F4/80−; Permeabilized surface-stained PCs were intracellularly stained with GT8, KO11 and IgG1 probes. Antibody stains were all diluted 100× into the cell sample. Cells were analyzed using the LSR II (BD Biosciences), the Aurora (Cytek), or sorted using a BD FACSAria (BD Biosciences) at the Scripps Flow core. FlowJo v10.7 was used to analyze all flow cytometry data and to generate figures.

**Allelic inclusion**. To assess the surface expression of endogenous LCs in engineered VRC01+ cells, B cells from knock-in mice expressing LCs with only human κ constant chains[33] were purified, activated, and engineered using the *H-targeting* strategy. Fc Blocker (homemade mAb 2.4g2) was added to single-cell suspensions at 0.5 mg per $10^6$ cells before antibody staining. The engineered B cells were stained with anti-mouse Kappa (BD, #561353), anti-human Kappa (Biolegend, #316506), anti-mouse Lambda (Biolegend, #407308), eOD-GT8-AF647 tetramer and analyzed by FACS.

**Peripheral tolerance**. Eight million B cells isolated from the spleens of CD45.2 congenic mice using negative selection (Miltenyi, #130-090-862) were either transferred directly or after 24 h of ex vivo LPS-activation into CD45.1 pAlb and WT recipient mice at day 0. Before adoptive transfer, the frequency of B cells carrying κ LC were determined using an AF647-labeled anti-κ antibody (homemade) by flow cytometry. The κ frequency of CD45.2 (donor) from splenic B cells of recipient mice were analyzed successively on day 1, 8, 15, and 28 post transfer.

**Total response ELISA**. 384-well ELISA plates (Corning, #3700) were initially coated with 12.4 μl/well streptavidin (Jackson Immuno Research Labs, #016-000-084) at 2 μg/ml diluted in PBS and incubated at 4 °C overnight. Plates were washed 3× with 100 μl/well PBS containing 0.05% Tween (PBS-T) and blocked with 40 μl/well PBS + 3% BSA at RT for 1 h before the addition of 12.4 μl/well biotin-labeled BG505 SOSIP (produced in house, Pugach et al., *Journal of Virology*, 2015) at 2 μg/ml diluted in PBS-T and 1% BSA. Mouse sera were serially diluted (3×) using PBS-T and 1% BSA starting at 1:10 and 12.4 μl/well incubated at room temperature (RT) for 1 h. After washing (as above), 12.4 μl/well alkaline phosphatase-conjugated goat anti-mouse IgG (H + L) (Jackson Immuno Research Labs, #115-055-146) diluted 1:5000 in PBS-T, 1% BSA was added and incubated for 30 min at RT. Plates were washed and p-nitrophenyl phosphate (pNPP) substrate (Sigma Aldrich, #S0942) dissolved to 1 mg/mL in substrate buffer (10 mM $MgCl_2$ with 80 mM $Na_2CO_3$ and 15 mM $NaN_3$, pH 9.8), was added at 12.4 μl/well to visualize the binding of antigen-specific mouse IgG. Optical density (OD) at 405 nm was read on a SpectraMax Plus (Molecular Devices) plate reader, allowing the same amount of development time for each plate. $EC_{50}$ values were generated by fitting curves to plots of Absorbance values vs. the log of the serum dilution for each sample.

**Anti-P2A ELISA**. 384-well plates were pre-coated overnight at 4 °C with 12.4 μl/well, 2 μg/ml BG505 SOSIP 2JD6 nanoparticle produced in house[34]. Plates were washed and blocked with 40 μl/well of PBS supplemented with 3% BSA at RT for 1 h and washed again. Mouse serum samples serially diluted (2×) with PBS-T and 1% BSA starting at 1:10 dilution were added (12.4 μl/well) and incubated at RT for 1 h. A P2A-LC tagged mouse VRC01 IgG monoclonal antibody standard starting at 1 μg/mL was also added (12.4 μl), serially diluted (2×), and incubated at RT for 1 h. Plates were washed and incubated with 12.4 μl/well biotin-labeled anti-2A peptide (3H4) mouse antibody (NovusBio, #NBP2-59627) at 1 μg/ml in PBS-T and 1% BSA. Plates were washed and the captured complex incubated with 12.4 μl/well alkaline phosphatase-conjugated Streptavidin (Jackson Immuno Research Labs, #016-050-084) at 1:3000 dilution in PBS-T and 1%BSA at RT for 1 h. Plates were washed and pNPP substrate was added as above. All plates developed for the same period of time before being read at 405 nm as above. P2A titers were reported as area under the curve for plots of Absorbance vs. log dilution factor for each sample. When maximum Abs 405 nm values were above 1.5, P2A-antibody quantities in serum samples were calculated by multiplying the sample EC50 with that of the mouse-VRC01 standard.

**Mouse VRC01−P2A IgG preparation**. The VRC01 LC-P2A-HC construct was PCR-amplified from engineered C57BL/6J mouse cDNA and cloned between the promoter and constant regions of a mouse IgG2b HC expression vector (InvivoGen, #pfuse-mchg2b) using Gibson Assembly. A leader sequence (5′-ATGGGA TGG TCATGTATCATCCTTTTTCTAGTAGCAACTGCAACCGGTGTACATT CA-3′) was also incorporated 5′ to the LC region. Transient expression of the P2A-VRC01 antibody was accomplished through transfection into FreeStyle™ 293-F cells following manufacturer's guidelines. The supernatant was harvested 5 days post transfection and IgG was affinity purified using Protein A Sepharose (GE Healthcare, #17−5280-02). SDS-PAGE was used to confirm the purity of IgG and adequate cleavage of the P2A peptide. The activity of the antibody was comparable to the human VRC01 IgG monoclonal antibody in virus neutralization assays.

**Competition ELISA**. 384-well plates were pre-coated overnight at 4 °C with 12.4 μl of BG505 SOSIP 2JD6 at 2 μg/ml diluted in PBS. After washing and blocking, 12.4 μl/well of mouse sera serially diluted (2×) in PBS-T and 1% BSA (starting at 1:10) was added to plates and incubated at RT for 1 h. 12.4 μl/well of human VRC01 monoclonal antibody at 200 ng/ml was then added directly to the diluted sera. Plates were incubated for another hour after mixing and non-binding antibodies were washed away. Captured antibodies were detected with alkaline phosphatase-conjugated goat anti-human IgG, Fcγ fragment specific (Jackson Immuno Research Labs, #109-055-098) diluted 1:5000 in PBS-T and 1% BSA. Plates were incubated at RT for 1 h before the addition of pNPP substrate as

described above. OD values at 405 nm were read, and standard curves were generated. Absorbance values were transformed by taking the absolute value of the (sample Abs 405 nm minus the Abs 405 nm of non-competing negative control wells). The area under the curve was then generated from Abs vs log serum dilution plots.

**Neutralization assay**. Under sterile BSL2/3 conditions, PSG3[35] plasmid was co-transfected into 293T cells along with various HIV envelope plasmids[31] using Lipofectamine 2000 transfection reagent (ThermoFisher Scientific, #11668019) to produce single-round of infection competent pseudo-viruses representing multiple clades of HIV. 293T cells were plated in advance overnight with DMEM medium + 10% FBS + 1% Pen/Strep + 1% L-glutamine. Transfection was done with Opti-MEM transfection medium (Gibco, #31985) using Lipofectamine 2000. Fresh medium was added 12 h after transfection. Supernatants containing the viruses were harvested 72 h later. In sterile 96-well plates, 20 μl of virus was immediately mixed with 20 μl of serially diluted (3×) purified IgG from mouse sera (starting at 400 μg/ml) and incubated for one hour at 37 °C to allow for antibody neutralization of the pseudoviruses. 5000 TZM-bl cells/well (in 40 μl of media containing 100 μg/ml Dextran) were directly added to the antibody virus mixture. Plates were incubated at 37 °C for 48 h. Following the infection, TZM-bl cells were lysed using 1× luciferase lysis buffer (25 mM Gly-Gly pH 7.8, 15 mM MgSO₄, 4 mM EGTA, 1% Triton X-100). Neutralizing ability disproportionate with luciferase intensity was then read on a Biotek Synergy 2 (Biotek) with luciferase substrate according to the manufacturer's instructions (Promega, #PR-E2620).

**Engineered B cell immunoglobulin repertoire sequencing and analysis**. Cells were released from mouse spleen and lymph nodes by pressing these tissues through a 0.2 μm cell strainer using a rubber syringe plunger and rinsing them into a 50 ml falcon tube using MACS buffer (PBS + 2%FBS + 2 mM EDTA). Bone marrow was obtained by crushing tissue-free tibia and femur bones with a mortar and pedestal and rinsing the released cells through a 0.2 μm cell strainer into a 50 ml falcon tube. Cells were pelleted by centrifugation ($600 \times g$ for 6 min). RBCs were disrupted by resuspending the cell pellets in 10 ml of RBC lysis buffer (155 mM $NH_4Cl$ + 12 mM $NaHCO_3$ + 0.1 mM EDTA) for 3 min at RT. Cells were then diluted to 50 ml with MACS buffer and pelleted. The cells were washed with 50 ml of MACS buffer and cell numbers determined using a Coulter Particle Counter. We routinely obtained ~1–1.4 × $10^8$ cells from the spleen and lymph node samples, and roughly 6–9 × $10^7$ cells from the marrow. Cells were pelleted and resuspended in 100 μl MACS buffer/$10^7$cells. 0.5 μg of FcR blocking reagent was added per $10^6$ cells and the mixture incubated for 10 min at RT. $10^6$ cells were kept for FACS analysis. The remaining cells were then labeled with biotinylated anti-CD45.2 Ab (Clone 104, Biolegend, #109804) at a 1:100 dilution for 30 min. The cells were washed twice with 1–2 ml MACS buffer per $10^7$ cells. Cell pellets were resuspended in 70 μl MACS buffer and 20 μl of anti-biotin microbeads (Miltenyi, #130-090-485)/$10^7$ cells. The suspension was well mixed by pipetting and incubated for 15 min at 4 °C. The cells were washed by adding 1–2 ml of MACS buffer per $10^7$ cells and pelleted. Cells were then resuspended in 500 μl of MACS buffer and pipetted onto a MACS buffer rinsed LS column (Miltenyi, #130-042-401) placed in a magnet field to retain labeled cells on the column. The column was rinsed 3 × 3 ml MACS buffer. The column was then removed from the magnet and labeled cells eluted into a falcon tube with 7 ml MACS buffer pushed through the column with a plunger. These cells were then subjected to a second round of purification over a new LS column. Cell numbers were measured and cells pelleted for mRNA purification. We routinely obtained ~4–6 × $10^6$ and 2–4 × $10^6$ cells for spleen/lymph node or bone marrow-derived cells respectively. $10^5$ cells were kept for FACS analysis. mRNA was purified using the Qiagen RNeasy micro kit (Qiagen, #74004) according to the manufacturer's instructions. mRNA was immediately subjected to reverse transcription using the Superscript III first-strand synthesis kit (Thermo) to generate cDNA following the manufacturer's instructions. For the RT-PCR we used either a VRC01-specific primer (5′-CCATCTCA TCCCTGCGTGTCTCCGAC NNNNNNNN GATGAGACGATGACCG -3′) or a mixture of isotype (IgM, IgA, IgE, IgD & IgG) specific primers (PID_mIgM- CCA TCTCATCCCTGCGTGTCTCCGAC NNNNNNNN CTGGATGA CTTCAGT GTTGT; PID_mIgA- CCATCTCATCCCTGCGTGTCTCCGAC NNNNNNNN CCAGGT CACATTCATCGTG; PID_mIgE- CCATCTCATCCCTGCGTGTCT CCGAC NNNNNNNN GTTCA CGTGCTCATGTTC; PID_mIgD− CCATCTCA TCCCTGCGTGTCTCCGAC NNNNNNNN GCCAT TTCTCATTTCAGAGG; PID_mIgG12- CCATCTCATCCCTGCGTGTCTCCGAC NNNNNNNNN KK ACAGTCACGGAGCTGCT; PID_mIgG3− CCATCTCATCCCTGCGTGTCTCC GAC NNNNNNNN GTACAGTCACCAAGCTGCT) containing unique primer IDs (PID) and a primer landing site. All primers for cDNA synthesis were HPLC purified. The PID were tagging each RNA template with a unique 8−nucleotide-long identifier, which allowed us to group amplified sequences for each individual template. The primer landing site was used for the reverse primer binding site of the subsequent hemi-nested PCRs. The resulting cDNA was purified using AMpureXP beads (Beckman Coulter, #A63882) at a volume ratio of 1:1. The purified cDNA was amplified and barcoded during a hemi-nested PCR using the forward primers VRC01_L1-F (5′-GGATTTTCATGTGCAGATTTTCAGCTT CATGC-3′) for the first round and the primer VRC01_Junc-F (5′-CAGTGTCACA GTCATATTGTCCAGTGG-3′) for the 2nd round PCR in combination with the

reverse primer PID-R (5′-ATCCCTGCGTGTCTCCGAC -3′), that attached to the primer landing site. Successful amplification was confirmed on a 0.7% agarose gel and amplicons were barcoded during an additional second round PCR using a barcoded second round primer set. The final barcoded amplicons were quantified using the Tapestation D5000 ScreenTape System (Agilent) and pooled at equimolar ratios. Library preparation and sequencing of SMRTbell template libraries of ~1.5 kb insert size were performed according to the manufacturer's instructions on a PacBio instrument (Pacific Biosciences) at the Salk Institute. Data were processed and visualized using a pipeline built for this sequencing protocol in the Julia language, using NextGenSeqUtils[36]. In brief, samples were first demultiplexed and oriented using demux_dict(), and reads were dereplicated. Sequences were collapsed by primer ID (UMI), and if sequences with the same PID differed, either due to sequencing error or PID "clashes"[37] we retained the most frequent variant. This gives us a dataset at the "transcript level". To exclude PacBio or RT indel errors, we discard reads with any indel variation relative to the engineered reference. For isotype mixture RT sequencing, we perform isotype calls on transcripts by matching the sequence immediately 3′ of the engineered polypeptide against isotype references extracted from the CH1 IMGT database, discarding transcripts that do not match, or match ambiguously. We then collapse "transcripts" into "variants" (sequences with 100% identical nucleotides over the polypeptide region, and, in the isotype RT datasets, identical isotype calls), retaining a transcript count for each variant. These variants, and their associated frequencies, were used in all sequence analyses (always including the variant frequency when counting any mutations). For phylogenetic analysis, but nowhere else, we additionally collapse any "singleton" variants (where frequency = 1) into larger variants of the same isotype that are just one nucleotide distance from them (we retain singletons that are >1 nucleotide from their nearest variant). This allows for compact display of phylogenies. Maximum likelihood phylogenetic trees were inferred using FastTree2[38], and visualized using FigTree (http://tree.bio.ed.ac.uk/software/figtree/). Curated data were also analyzed using Geneious 11.0.5.

**Statistical analysis**. Statistical analysis used an unpaired two-tailed $T$ test (Prism, Graphpad). Correlation between EC50 of P2A titers and the relative concentration of VRC01 was calculated using the Pearson correlation coefficient with linear regression (Prism, Graphpad). $*P < 0.05$, $**P < 0.01$, $***P < 0.001$, $****P < 0.0001$.

**Reporting summary**. Further information on research design is available in the Nature Research Reporting Summary linked to this article.

## Data availability

Annotated donor DNA sequences are available in GenBank under accession codes MT789856, MT789857, and MT789858. Raw flow cytometry data that support the findings of this study are available from the corresponding author (J.E.V.) upon reasonable request. Raw PacBio data (Fig. 4, Supplementary Figs. 6 and 7) are available from the corresponding author (B.M.). All other data are available in the article and supplementary files or from the corresponding author J.E.V. upon reasonable request. Source data are provided with this paper.

## Code availability

NGS analysis code and intermediate data processing steps are available in GitHub (https://github.com/MurrellGroup/EngineeredBCRseq/).

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

## Acknowledgements

We thank Nicolle Jigarjian and staff at the Scripps Department of Animal Resources for the ongoing care of experimental mice, Darrell Irvine at MIT for providing IscoMPLA adjuvant, and Christina Corbaci for assistance making figures. This work was supported by the Bill and Melinda Gates Foundation (grant number OPP1183956 to J.E.V.) and by the National Institutes of Health (5R01DE025167−05 to D.R.B. and R01AI128836 and R01AI073148 to D.N.).

## Author contributions

D.H. and J.T.T., A.O., and M.T. developed methods and reagents for B cell engineering, carried out adoptive transfer and immunization experiments, collected mouse serum and tissues, performed ELISA, FACS analysis, virus neutralization assays, and produced immunogens and antibodies. J.E.V, T.V., M.V.G., and B.M. generated engineered repertoire libraries, sequenced the libraries, analyzed the sequence datasets, generated figures and edited the manuscript. J.A., L.C., T.R.B., K.S., W.L., E.L., and A.G.M. assisted with methods development, experiments or animal care. T.S. and W.S. provided HIV vaccine immunogens and FACS probes. J.E.V., D.H., D.N., and B.M. designed the experiments. J.E.V., D.N., D.H., D.R.B., and B.M. analyzed the data and wrote the manuscript. All authors edited the manuscript.

## Competing interests

The authors declare no competing interests.
