## [Peer Review File · Nature Communications]

REVIEWER COMMENTS

Reviewer #1 (Remarks to the Author):

The manuscript by Huang et al. describes the engineering of B cell to express an anti-HIV broadly neutralizing antibody (bNAb) in vivo. The unique aspect of this study is the insertion, via CRISPR/Cas9, of the anti-HIV VRC01 clone bNAb genes (encoding the heavy and light chains) into mouse B cell receptor (BCR) loci. This enables VRC01 to potentially replace the endogenous BCR, undergo antibody class switching, undergo somatic hypermutation, and result in B cell expansion and maturation into memory and plasma cells. They attempted two different approaches, insertion of the VRC01 heavy and light chains, separated by the P2A sequence, into the mouse heavy chain loci (termed H-targeted) and insertion of the VRC01 heavy chain into the mouse heavy chain locus and the VRC01 kappa chain into the mouse kappa chain locus (termed H+k targeted). Both approaches proved feasible, but the H+k targeted approach was relatively inefficient in dual gene modification and BCR expression. Following transfer of modified cells into mice and immunization, animals receiving H-targeted VRC01 elicited engineered antibody responses produced the largest levels of responses (mice receiving H+k targeted cells had very low level, if any, responses). The H-targeted animals demonstrated engineered B cell expansion/differentiation into memory and plasma cells, class switched antibody, and evidence of somatic hypermutation. Their results in the H-targeted approach largely support their conclusions. However, there are issues that could be addressed to strengthen this manuscript:

1. The lack of the H+k targeted approach to elicit significant responses is a concern. The endogenous kappa chain is likely expressed in the H-targeted strategy, which could lead to chain mispairing with the endogenously produced kappa chain and the production of self-reactive antibodies. The authors attempt to circumvent this issue in Figure 1d by showing that "auto-reactive" kappa chain expressing B cells can be deleted following their cell-processing procedures and transfer into pAlb mice (which contain a liver expressed single-chain antibody reactive to mouse Ig κ). This mechanism of B cell tolerance is different from potential self-reactivity from mis-paired BCRs and does not indicate that self-reactive B cells produced from the insertion of engineered BCRs would also be deleted. In context, the most interesting part of this study is that the modification of the mouse heavy chain region with an HIV-specific bNAb results in production of an antigen specific BCR that can potentially evolve with an immune-selected virus. However, the presence of the endogenous kappa chain and potential self-reactivity is an overall issue with the approach. This could minimally be better explained in the text in how future studies will examine this. Optimally, this manuscript would be strengthened by data showing whether or not the endogenous kappa is expressed in H-targeted cells and if mispaired antibodies are produced. It is also unknown whether the differences in the human antibody (VRC01) and mouse sequences would favor the sole production of the human antibody. This could be addressed.
2. Is the lack of response in the H+k targeted approach due to inefficiency of the gene targeting/modifications? The reasons that this may not have elicited a strong response in vivo is not clear in the text.
3. The authors use cellular phenotype to identify germinal center B cells ex vivo. Histologic analysis showing the presence of modified cells in germinal centers would strengthen their results as the cell processing procedures could produce these cellular activation markers.

Reviewer #2 (Remarks to the Author):

The manuscript by Huang et al. titled "Vaccine Elicitation of HIV Broadly Neutralizing Antibodies from Engineered B cells" explored a strategy of engineering B cells encoding HIV bnAb as vaccine or means of HIV functional cure that enables durable elicitation of corresponding protective antibodies in wild-type black 6 mice. The authors employed CRISPR-Cas9 technology to insert the

HIV bnAb VRC01 heavy and light chain variable regions into the J4 site of mouse HC locus of donor mouse primary splenic B cells that have been activated by LPS, followed by transferring into host mice. These transferred engineered VRC01-encoding B cells displayed a germinal center (GC)-independent (CD73-) memory phenotype, and survived well in the host mice. The host mice were then immunized with nanoparticles bearing HIV Env BG505 SOSIP trimer, MD39-ferritin, for a few times that resulted in expansion of VRC01-encoding memory and plasma B cells in spleen and long-lived plasma cells in the bone marrow. VRC01-encoding germinal center B cells are also observed. The sera of the host mice demonstrated antigen binding specificity similar to VRC01. The purified serum IgG of the host mice showed moderate to strong neutralization capacities to selected tier 2 viruses, consistent with the bnAb feature of VRC01. Additionally, typical germinal center antigen-specific B cells were observed in the host mice. Somatic hypermutations were observed from the BCRs of transferred engineered B cells post immunizations. This proof-of-concept study demonstrated the feasibility of transforming bnAb passive immunization into active immunization that could afford durable and inducible protection by engineering B cells carrying bnAb-encoding BCRs, which may open new avenues for HIV prevention and functional cure. The novel approach to manipulate BCRs of primary B cells by CRISPR-Cas9 technology and to test the in vivo function in small animal model will serve as template for the field. Some suggestions are listed as below to improve the clarity of the manuscript.

Major points

- (1). The authors may consider highlighting the purpose of using LPS to activate B cells prior to the BCR engineering, which apparently is to push more transferred cells towards memory B cell status. What is the possible mechanism underlying this? Would this cause potential somatic hypermutations? Such as the kappa chain constant region revealed by sequencing in Fig.4?
- (2). The most promising animals are H5 & H6. The quantification of VRC01 in total IgG is 1-10% (w33, Fig.2j). Accordingly, in the virus neutralization assay, while the total IgG concentration was set at 200 ug/mL, the concentration of VRC01 should be 2-20 ug/mL in this assay. However, the percentage of entry blocking by total IgG for BGN (BG505) is below or marginally around 50%. According to previous studies, VRC01 IC50 value for BG505 is below 0.1 ug/ml. Why there is a huge discrepancy for the BGN-neutralization potency between the mouse-produced VRC01 and the bnAb VRC01? Was it caused by assay variation or other factors (e.g. the use of mouse constant region in this study)? Was a mouse VRC01 IgG containing P2A peptide used as control in this BGN neut. assay? Some clarification would be helpful.
- (3). Fig.3, IgG1 is used to gate GC, memory, and plasma cells. What about the other subclass? Is IgG1 the most representative subclass in the B cell response to the immunogen MD-ferritin, or it is a feature of LPS as adjuvant?
- (4). Fig.4 Sequencing of the BCRs. A lot of mutations occur within the kappa constant gene, which is not normally in the path of the mutator. What will happen to the encoded antibody with these mutations? Are these mutations at the interface of CK and CH1 domains? Would this affect the heterodimer formation between the kappa chain and the heavy chain, which will lead to the reduced production of functional antibody? Some structural analysis may be helpful (optional).

Minor point:

Fig.2a, Y-axis, MCs of donor B cells (%), please specify what is MCs (memory B cells sometimes are referred to as MBC, sometimes as MC).

Reviewer #3 (Remarks to the Author):

Huang et al. here describe a CRISPR/Cas-based approach for engineering primary mouse B cells to express Broadly Neutralizing Antibodies from the endogenous BCR locus. Adoptive transfer into immunocompetent mice results in durable B cells with somatic hypermutation after repeated immunization.

As my primary expertise is genome engineering, I will mainly comment on these aspects of the manuscript.

(1) There seems to be some overlap to the authors' previous publication in eLife (PMID: 30648968). The previous strategy, the results, as well as the new findings and contributions (novelty) of the new manuscript should be more clearly described.

(2) In the flow staining presented in Figure 1c, it is not clear what is going on. In the text, it is stated that VRC01 expression is measured "using flow cytometry probes specific for this antibody, eOD-GT8 and KO11". It is unclear from their names what these probes are and what they bind. Please explain. Why does Fig. 1c display the use of two eOD-GT8 fluorophores? Please show the full gating scheme for this analysis.

(3) Plasmid-based DNA donor is used for HDR. Plasmid is known to be highly toxic to primary cells so a thorough analysis of toxicity is warranted. This includes viability analysis and absolute cell counts >2 days post delivery including control samples that are non-electroporated, mock-electroporated without reagents, electroporated with Cas9 reagents only, and plasmid only.

(4) Detailed analyses of the genomic outcome following gene editing are lacking. INDEL rates are not quantified, and the functional outcome of NHEJ events are not described. For scientist that are little familiar with this genomic locus, are the sgRNAs targeting coding regions and if so, what would the functional outcome be for mutagenic NHEJ. The use of two sgRNAs can cause translocations (assuming they target different chromosomes). I acknowledge the study is performed in mice and the reagents are not directly applicable to the human genome (please specify if they are), so a quantification of translocations is not necessary, but the phenomenon should be discussed.

(5) The authors state that some mutations were observed in the VRC01 indicative of somatic mutation and affinity maturation. Did the authors perform NGS on the plasmid DNA before genetic engineering or the B cell population immediately after genetic engineering to rule out (or at least quantify levels of) pre-existing mutations? This would further substantiate the claim that somatic hypermutation occurs.

REVIEWER COMMENTS

Reviewer #1 (Remarks to the Author):

The manuscript by Huang et al. describes the engineering of B cell to express an anti-HIV broadly neutralizing antibody (bNAb) *in vivo*. The unique aspect of this study is the insertion, via CRISPR/Cas9, of the anti-HIV VRC01 clone bNAb genes (encoding the heavy and light chains) into mouse B cell receptor (BCR) loci. This enables VRC01 to potentially replace the endogenous BCR, undergo antibody class switching, undergo somatic hypermutation, and result in B cell expansion and maturation into memory and plasma cells. They attempted two different approaches, insertion of the VRC01 heavy and light chains, separated by the P2A sequence, into the mouse heavy chain loci (termed H-targeted) and insertion of the VRC01 heavy chain into the mouse heavy chain locus and the VRC01 kappa chain into the mouse kappa chain locus (termed H+k targeted). Both approaches proved feasible, but the H+k targeted approach was relatively inefficient in dual gene modification and BCR expression. Following transfer of modified cells into mice and immunization, animals receiving H-targeted VRC01 elicited engineered antibody responses produced the largest levels of responses (mice receiving H+k targeted cells had very low level, if any, responses). The H-targeted animals demonstrated engineered B cell expansion/differentiation into memory and plasma cells, class switched antibody, and evidence of somatic hypermutation. Their results in the H-targeted approach largely support their conclusions. However, there are issues that could be addressed to strengthen this manuscript:

1. The lack of the H+k targeted approach to elicit significant responses is a concern. The endogenous kappa chain is likely expressed in the H-targeted strategy, which could lead to chain mispairing with the endogenously produced kappa chain and the production of self-reactive antibodies. The authors attempt to circumvent this issue in Figure 1d by showing that "auto-reactive" kappa chain expressing B cells can be deleted following their cell-processing procedures and transfer into pAlb mice (which contain a liver expressed single-chain antibody reactive to mouse Igk). This mechanism of B cell tolerance is different from potential self-reactivity from mis-paired BCRs and does not indicate that self-reactive B cells produced from the insertion of engineered BCRs would also be deleted. In context, the most interesting part of this study is that the modification of the mouse heavy chain region with an HIV-specific bNAb results in production of an antigen specific BCR that can potentially evolve with an immune-selected virus. However, the presence of the endogenous kappa chain and potential self-reactivity is an overall issue with the approach. This could minimally be better explained in the text in how future studies will examine this. Optimally, this manuscript would be strengthened by data showing whether or not the endogenous kappa is expressed in H-targeted cells and if mispaired antibodies are produced. It is also unknown whether the differences in the human antibody (VRC01) and mouse sequences would favor the sole production of the human antibody. This could be addressed.

There are very few publications that describe BCR engineered B cells at this time, however none of these studies give any consideration to the idea that peripheral tolerance mechanisms *in vivo* could play a vital role in inactivating autoreactive cells that may be generated by engineering methods. This perspective is important because it highlights the fact that attempts to reduce the expression of autoreactive BCRs in engineered cells is more likely to have an impact on overall vaccine efficacy than on preventing the generation of autoreactive B cell responses. *i.e.* fewer autoreactive engineered cells will translate into more cells which are able to expand and mature in response to vaccination in the context of a normal immune system. Furthermore, presenting the fate of autoreactive engineered cells as we have done also hints at potential applications for 'Engineered B cell Vaccines' in this first proof-of-concept paper. For example, it shows us that this system would not work for the long-term expression of mAbs targeting self-antigens such as those often used in cancer immunotherapy. Another argument against biasing the BCR engineering literature against autoreactivity comes from the idea that autoreactive cells can eliminate their autoreactivity by mutation- a phenomenon that Goodnow calls 'redemption' (PMID: 31556969). Since the engineered B cells efficiently mutate in response to vaccination in this model, that mechanism could convert autoreactive cells into useful ones. For example, two out of every three indels generated in the endogenous LC gene during affinity maturation would eliminate the expression of the problematic endogenous chain via frameshift mutation. In this version of the paper, however, we have now characterized the fraction of *H-targeted*, VRC01 expressing B cells that present cell endogenous light chains on their surface as mis-paired antibodies, and which thus have the potential to be auto or polyreactive. We have accomplished this by performing *H-targeting* in primary B cells from knock-in mice which express only kappa chains which use human constant genes (<https://www.biorxiv.org/content/biorxiv/early/2020/02/27/2020.02.24.963629.full.pdf>). *H-targeted*

VRC01⁺ cells were stained with anti-human kappa and anti-mouse lambda FACS probes to detect cell native light chains that might present on the cell surface as mis-paired BCRs. Interestingly, the number of human kappa positive cells was significantly reduced on VRC01⁺ cells compared with mock-targeted cells (*Figure S2*). This suggests that, even though all engineered cells should express a native light chain in addition to the engineered VRC01 mouse kappa chain, a significant fraction of the engineered cells may have endogenous light chains that cannot compete with the engineered mouse VRC01 kappa chain for pairing with the VRC01 heavy chain. The *H+K targeting* strategy was actually developed, in part, to reduce the fraction of autoreactive engineered cells, because endogenous kappa chain expression should be mostly interrupted when the new light chain is successfully incorporated (PMID: 30975893). We are currently exploring a variety of BCR engineering strategies that would result in less or no autoreactive B cells, but in order to keep a sharp focus and to be less speculative, we hope that this reviewer finds it acceptable that we do not discuss all of these in this proof-of-concept short report.

2. Is the lack of response in the H+k targeted approach due to inefficiency of the gene targeting/modifications? The reasons that this may not have elicited a strong response in vivo is not clear in the text.

Although the engineering efficiency of the *H+k targeting* approach is about 10-fold lower, equal numbers of engineered B cells (GT8++ /KO11-) for both H and H+k strategies were transferred to the recipient mice to normalize for gene targeting efficiency differences during the analysis of vaccine responses. We did not speculate on the reasons this strategy did not work, as there are a wide variety of possibilities that we are currently investigating. Ideally, we prefer to use this targeting strategy as: 1) fewer engineered cells would be autoreactive due to mis-paired BCRs as discussed above; 2) The light and heavy chains might perhaps be expressed more naturally as two separate transcripts spliced to endogenous cell constant genes and; 3) We would eliminate any potential problems that could be the result of somatic hypermutation of the VRC01 mouse kappa constant gene or expression of LCs with a C-term P2A tag. We now more clearly state that further investigations will be required to understand why *H+K targeted* cells failed to elicit VRC01 responses on line 132-3 (revised_manuscript.pdf), after presenting the serology results.

3. The authors use cellular phenotype to identify germinal center B cells ex vivo. Histologic analysis showing the presence of modified cells in germinal centers would strengthen their results as the cell processing procedures could produce these cellular activation markers.

We agree that we have provided proof that the B cells can acquire a germinal center B phenotype, but not that they reside in germinal centers. However, the hallmarks of germinal center responses which include BCR class switching and somatic hypermutations, have been clearly demonstrated 14 days post boost through flow cytometry and by next generation sequencing at the end of the experiment. We hope that these demonstrations satisfy this reviewer's concerns.

Reviewer #2 (Remarks to the Author):

The manuscript by Huang et al. titled "Vaccine Elicitation of HIV Broadly Neutralizing Antibodies from Engineered B cells" explored a strategy of engineering B cells encoding HIV bnAb as vaccine or means of HIV functional cure that enables durable elicitation of corresponding protective antibodies in wild-type black 6 mice. The authors employed CRISPR-Cas9 technology to insert the HIV bnAb VRC01 heavy and light chain variable regions into the J4 site of mouse HC locus of donor mouse primary splenic B cells that have been activated by LPS, followed by transferring into host mice. These transferred engineered VRC01-encoding B cells displayed a germinal center (GC)-independent (CD73-) memory phenotype and survived well in the host mice. The host mice were then immunized with nanoparticles bearing HIV Env BG505 SOSIP trimer, MD39-ferritin, for a few times that resulted in expansion of VRC01-encoding memory and plasma B cells in spleen and long-lived plasma cells in the bone marrow. VRC01-encoding germinal center B cells are also observed. The sera of the host mice demonstrated antigen binding specificity similar to VRC01. The purified serum IgG of the host mice showed moderate to strong neutralization capacities to selected tier 2 viruses, consistent with the bnAb feature of VRC01. Additionally, typical germinal center antigen-specific B cells were observed in the host mice. Somatic hypermutations were observed from the BCRs of transferred engineered B cells post immunizations. This proof-of-concept study demonstrated the feasibility of

transforming bnAb passive immunization into active immunization that could afford durable and inducible protection by engineering B cells carrying bnAb-encoding BCRs, which may open new avenues for HIV prevention and functional cure. The novel approach to manipulate BCRs of primary B cells by CRISPR-Cas9 technology and to test the *in vivo* function in small animal model will serve as template for the field. Some suggestions are listed as below to improve the clarity of the manuscript.

Major points

(1). The authors may consider highlighting the purpose of using LPS to activate B cells prior to the BCR engineering, which apparently is to push more transferred cells towards memory B cell status. What is the possible mechanism underlying this? Would this cause potential somatic hypermutations? Such as the kappa chain constant region revealed by sequencing in Fig.4?

The primary reason for using LPS to activate mouse primary B cells, was because this method gave us the best genome editing efficiencies (entry into cell cycle is necessary for gene targeting by homology directed repair). This is now more clearly explained in the text on lines 86-88, 99-101 (revised_manuscript.pdf). Maturation from naive to memory B cells *in vivo* after LPS-activation *in vitro* was unexpected and unintended. The mechanism of this phenotypic change is not understood and, though interesting, is beyond the scope of the current manuscript. It has been previously shown that primary B cells stimulated with LPS do not undergo somatic hypermutation *in vitro* (Ref, PMID: 28939756, 11160276, 3108398). Additionally, VRC01 IgM sequence have almost no mutations in them after vaccination. Because human cells cannot be activated with LPS, we do show that the engineering of cells and subsequent elicitation of VRC01 responses after boost can be achieved when alternative MyD88 inducers such as CpG DNA are used to activate cells (Table S1).

(2). The most promising animals are H5 & H6. The quantification of VRC01 in total IgG is 1-10% (w33, Fig.2j). Accordingly, in the virus neutralization assay, while the total IgG concentration was set at 200 ug/mL, the concentration of VRC01 should be 2-20 ug/mL in this assay. However, the percentage of entry blocking by total IgG for BGN (BG505) is below or marginally around 50%. According to previous studies, VRC01 IC50 value for BG505 is below 0.1 ug/ml. Why there is a huge discrepancy for the BGN-neutralization potency between the mouse-produced VRC01 and the bnAb VRC01? Was it caused by assay variation or other factors (e.g. the use of mouse constant region in this study)? Was a mouse VRC01 IgG containing P2A peptide used as control in this BGN neut. assay? Some clarification would be helpful.

We thank this reviewer for pointing out this important error in figure 2. The level of BG505 neutralization was actually 100% for the H-6 w33 sample. We have corrected this figure and have added two supplementary figures to show our P2A-ELISA (Figure S4) and neutralization results (Figure S5) in more detail. We hope we have now also better described the use of a recombinant VRC01 mouse IgG LC-P2A tagged standard for quantification of engineered antibody in the serum and as a fraction of the total serum IgG from the indicated samples. Figure S5 also compares neutralization curves for the human vs. P2A-tagged mouse VRC01 IgG standards for several viruses. These curves show that VRC01 neutralization IC50s are very minimally impacted when this antibody is expressed using mouse vs. human constant genes. We hope these changes improve the readability and clarity of our manuscript.

(3). Fig.3, IgG1 is used to gate GC, memory, and plasma cells. What about the other subclass? Is IgG1 the most representative subclass in the B cell response to the immunogen MD-ferritin, or it is a feature of LPS as adjuvant?

IgG1 was chosen because it is a dominant subclass and the anti-IgG1 probe used is particularly reliable and specific. We wished to avoid spurious cross reactivities associated with polyclonal anti-IgG antibodies and to limit the number of probes used in the FACS analysis. Class switch to the expected proportion of all other isotypes was detected by engineered repertoire sequencing in Figure 4.

(4). Fig.4 Sequencing of the BCRs. A lot of mutations occur within the kappa constant gene, which is not normally in the path of the mutator. What will happen to the encoded antibody with these mutations? Are these mutations at the interface of CK and CH1 domains? Would this affect the heterodimer formation between the kappa chain and the heavy chain, which will lead to the reduced production of functional antibody? Some structural analysis may be helpful (optional).

We now provide a brief summary of the location of kappa mutations lines 160-162 (revised_manuscript.pdf), and a new supplementary figure that makes it easier for the reader to see where coding mutations arise in the vaccinated engineered repertoire of a representative animal with reference to the structure (*Figure S7b*). As one might expect, most of these coding changes occur in solvent exposed loops that project from the Ig beta-barrel. Changes at these positions are likely to be inert. We noted the introduction of a predicted N-linked glycan site that could possibly function to improve the solubility, pairing or expression of the engineered antibody. We believe changes in this region of the engineered gene should either be inert or fortuitous if they are to be selected during competition in germinal center reactions. While we hope to study the phenomenon in much greater depth if *H-targeting* becomes embraced for the development of engineered B cell vaccines, we believe that a detailed study is beyond the scope of this paper which is showing for the first time, that durable bnAbs can be elicited from engineered cells through vaccination.

Minor point:

Fig.2a, Y-axis, MCs of donor B cells (%), please specify what is MCs (memory B cells sometimes are referred to as MBC, sometimes as MC).

Thank you for pointing out the inconsistency of labeling in figures. MCs means Memory B Cells. To keep consistency, MC is used to refer to memory B cells in all figures.

Reviewer #3 (Remarks to the Author):

Huang et al. here describe a CRISPR/Cas-based approach for engineering primary mouse B cells to express Broadly Neutralizing Antibodies from the endogenous BCR locus. Adoptive transfer into immunocompetent mice results in durable B cells with somatic hypermutation after repeated immunization.

As my primary expertise is genome engineering, I will mainly comment on these aspects of the manuscript.

(1) There seems to be some overlap to the authors' previous publication in eLife (PMID: 30648968). The previous strategy, the results, as well as the new findings and contributions (novelty) of the new manuscript should be more clearly described.

This report along with the accompanying manuscript by the group of Adi Barzel, are the first to show that engineered primary B cells are able to expand and undergo antigen-dependent maturation *in vivo* by way of signaling through their engineered B cell receptors. Our previous study analyzed B cell lines *in vitro* and demonstrated low level engineering of primary cells only. We uniquely show here that engineered cells can enter germinal centers, class switch, and differentiate into long-lived plasma cells and boostable memory in immunocompetent hosts in response to specific immunogens. Most importantly, we show that it is possible to generate durable HIV bnAb serum responses in an animal model that, like humans, is genetically restricted by its BCR repertoire with regards to its ability to generate HIV bnAbs through vaccination. These results encourage further exploration of engineered B cell vaccines as a strategy for durable elicitation of HIV bnAbs to protect against infection and as a contributor to a functional cure. We have made modifications to the manuscript to hopefully more effectively communicate the critical advances presented here.

(2) In the flow staining presented in Figure 1c, it is not clear what is going on. In the text, it is stated that VRC01 expression is measured "using flow cytometry probes specific for this antibody, eOD-GT8 and KO11". It is unclear from their names what these probes are and what they bind. Please explain. Why does Fig. 1c display the use of two eOD-GT8 fluorophores? Please show the full gating scheme for this analysis.

eOD-GT8 is an HIV envelope glycoprotein based recombinant protein with picomolar binding affinity for VRC01. KO11 is the same protein with a single amino acid substitution that knocks out VRC01 binding and is used to gate out B cells that bind off-target epitopes on eOD-GT8 in the flow cytometry analysis. This is now more clearly explained in the text lines 72-76, (revised_manuscript.pdf), and original references to the papers that describe these remain. A full gating strategy is presented as requested in *Figure S1*. Biotinylated eOD-GT8 and KO11 were pre-conjugated with streptavidin fluorophores before adding to the cells (as described in the methods). The two eOD-GT8 fluorophores were used to

exclude B cells recognizing the fluorophores themselves (*Figure S1* legend). The GT8⁺⁺/K011⁻ is the most conservative, and mock engineered samples do not contain cells in this gate.

(3) Plasmid-based DNA donor is used for HDR. Plasmid is known to be highly toxic to primary cells so a thorough analysis of toxicity is warranted. This includes viability analysis and absolute cell counts >2 days post-delivery including control samples that are non-electroporated, mock-electroporated without reagents, electroporated with Cas9 reagents only, and plasmid only.

We agree that plasmid-based donor DNA is quite toxic to primary cells. We now show live cell gates for the controls this reviewer has asked for in *Figure S1*. Despite this toxicity, we like to use the plasmid donor format to explore and develop engineering strategies in the wild type mouse model because it allows us to test a large number of donor DNA variants quickly and economically. We are also able to achieve engineering efficiencies using plasmid donors which compare with those achieved using the more conventional and less toxic adeno-associated virus (AAV) vector. We now also show engineering efficiency results using AAV-vectored delivery of single stranded donor DNA, to highlight that what we are doing can be translated to methods more widely accepted for clinical use (*Figure S1*).

(4) Detailed analyses of the genomic outcome following gene editing are lacking. INDEL rates are not quantified, and the functional outcome of NHEJ events are not described. For scientist that are little familiar with this genomic locus, are the sgRNAs targeting coding regions and if so, what would the functional outcome be for mutagenic NHEJ. The use of two sgRNAs can cause translocations (assuming they target different chromosomes). I acknowledge the study is performed in mice and the reagents are not directly applicable to the human genome (please specify if they are), so a quantification of translocations is not necessary, but the phenomenon should be discussed.

We now show cutting efficiencies for the *IgH-J4* and *IgK-J5* targeting RNPs used in this study in *Figure S1*. These were quantified by electroporating RNPs into cells individually without donor DNA and harvesting gDNA 3 days later. Primers were then used to generate two amplicons by PCR containing either cut site which were then assessed for indels using TIDE. The reviewer is correct that we did not engage in a discussion of off-target effects in this paper for several reasons. The main reason is that, in this early stage, we are exploring a variety of genome engineering strategies in the mouse model. Our intention is to do a deep analysis of only the most promising strategies from this model that we would like to advance. For example, at this stage we have advanced the *H-targeting* approach for use in rhesus and human B cells. RNPs must be redesigned for these genomes and the donors modified to accommodate the new cut site, homology regions etc. We are opting to do the deepest analysis of off-target effects in human cells if results from the rhesus macaque model are also promising. In the mouse model presented here, RNPs are targeting coding regions in the J genes. This is now clarified in the text and in *Figure 1* (these genes are only a few amino acids in length) and gRNA sequences are given in the methods. Most off-target repair of RNP cuts by NHEJ will either be inert or generate BCR knockouts which will lead to cell apoptosis. Chromosomal translocations involving telomeric ends of *IgH* (mouse chr12) and *IgK* (mouse chr6) RNP cut sites would lead to a loss of BCR expression and apoptosis of these B cells.

(5) The authors state that some mutations were observed in the VRC01 indicative of somatic mutation and affinity maturation. Did the authors perform NGS on the plasmid DNA before genetic engineering or the B cell population immediately after genetic engineering to rule out (or at least quantify levels of) pre-existing mutations? This would further substantiate the claim that somatic hypermutation occurs.

To show that the VRC01 gene mutations sequenced by NGS were indeed the result of somatic hypermutation (SHM), we now include another panel in *figure 4* (b) and a new *Supplementary Figure 6* depicting mutation frequency by isotype. Each colored dot represents an error corrected VRC01 sequence from immunized mice with *H-targeted* cells plotted by % n.t. mutation (left Y-axis). The IgM and IgG3 sequences, while few, were unmutated in contrast to IgG1, IgG2B, IgG2A/C, IgE and IgA. If the mutations observed had been introduced during the engineering step, we would expect the mutation frequencies to be the same for all isotypes. The *Figure S6* bar graphs show the fraction of sequences by isotype which had >0.1% mutation frequency. While this frequency is less than 5% for IgM and IgG3, it is greater than 60% for all other isotypes with the differences being highly significant in a Mann-Whitney test.

REVIEWERS' COMMENTS:

Reviewer #1 (Remarks to the Author):

The authors have submitted an improved revised manuscript that adequately addresses the reviewers' concerns. For this reviewer, the issue of the potential generation of self-reactive antibodies involving the kappa chain in vivo remains to be determined; however, within the scope of this study their explanations to this and the other concerns are sufficient. This study is highly unique serving as a proof of concept.

Reviewer #2 (Remarks to the Author):

The authors have largely addressed the concerns and improved the clarity of the manuscript.

Reviewer #3 (Remarks to the Author):

(1) No further comments

(2) No further comments

(3) The data showing lower toxicity when using AAV for donor delivery should be mentioned in the main text. The same goes for the rationale (presented in the rebuttal) for choosing DNA plasmid donor over AAV.

(4) The new data presenting INDEL rates following RNP delivery should be described in the main text. So should the discussion presented in the rebuttal regarding the consequences of INDELS and potential translocations. These are important points when proceeding with clinical translation.

(5) No further comments

REVIEWERS' COMMENTS:

Reviewer #1 (Remarks to the Author):

The authors have submitted an improved revised manuscript that adequately addresses the reviewers' concerns. For this reviewer, the issue of the potential generation of self-reactive antibodies involving the kappa chain in vivo remains to be determined; however, within the scope of this study their explanations to this and the other concerns 299 are sufficient. This study is highly unique serving as a proof of concept.

We thank this reviewer for his/her careful consideration.

Reviewer #2 (Remarks to the Author):

The authors have largely addressed the concerns and improved the clarity of the manuscript.

We thank this reviewer for his/her careful consideration.

Reviewer #3 (Remarks to the Author):

We thank this reviewer for his/her careful consideration.

(1) No further comments

(2) No further comments

(3) The data showing lower toxicity when using AAV for donor delivery should be mentioned in the main text. The same goes for the rationale (presented in the rebuttal) for choosing DNA plasmid donor over AAV.

(4) The new data presenting INDEL rates following RNP delivery should be described in the main text. So should the

discussion presented in the rebuttal regarding the consequences of INDELS and potential translocations. These are important points when proceeding with clinical translation.

We have mentioned points 3) and 4) in the main text.

(5) No further comments